Corrected: Author correction

# Low frequency transcranial electrical stimulation does not entrain sleep rhythms measured by human intracranial recordings

Belen Lafon [1], Simon Henin [2,3], Yu Huang [1], Daniel Friedman [2,3], Lucia Melloni [2,3,4], Thomas Thesen [3,5], Werner Doyle[2,6], György Buzsáki [3,7], Orrin Devinsky [2,3], Lucas C. Parra [1] & Anli A. Liu [2,3]

Transcranial electrical stimulation has widespread clinical and research applications, yet its effect on ongoing neural activity in humans is not well established. Previous reports argue that transcranial alternating current stimulation (tACS) can entrain and enhance neural rhythms related to memory, but the evidence from non-invasive recordings has remained inconclusive. Here, we measure endogenous spindle and theta activity intracranially in humans during low-frequency tACS and find no stable entrainment of spindle power during non-REM sleep, nor of theta power during resting wakefulness. As positive controls, we find robust entrainment of spindle activity to endogenous slow-wave activity in 66% of electrodes as well as entrainment to rhythmic noise-burst acoustic stimulation in 14% of electrodes. We conclude that low-frequency tACS at common stimulation intensities neither acutely modulates spindle activity during sleep nor theta activity during waking rest, likely because of the attenuated electrical fields reaching the cortical surface.

[1] Department of Biomedical Engineering, City College of New York, 160 Convent Ave, New York, NY 10031, USA. [2] New York University Comprehensive Epilepsy Center, 223 East 34th Street, New York, NY 10016, USA. [3] Department of Neurology, New York University School of Medicine, 240 East 38th St, 20th Floor, New York, NY 10016, USA. [4] Department of Neuroscience, Max Planck Institute for Empirical Aesthetics, Gruneburgweg 14, 60322 Frankfurt am Main, Germany. [5] Department of Physiology and Neuroscience, St. George's University, St. George's, Grenada. [6] Department of Neurosurgery NYU School of Medicine, 530 1st Avenue, Suite 7W, New York, NY 10016, USA. [7] New York University Neuroscience Institute, 450 East 29th St, New York, NY 10016, USA. Belen Lafon and Simon Henin contributed equally to this work. Lucas C. Parra and Anli Liu jointly supervised this work. Correspondence and requests for materials should be addressed to A.Liu. (email: anli.liu@nyumc.org)

The therapeutic potential of transcranial electrical stimulation (TES) has been examined in over 70 diverse conditions, including major depression, epilepsy, pain, stroke rehabilitation, Parkinson's disease, and tinnitus[1]. TES is motivated by the well-established biophysical observation that externally applied electric fields can affect neuronal excitability[2–4]. However, behavioral effects in human are often weak and difficult to replicate, which underscores a lack of basic mechanistic understanding of how applied electrical fields interact with brain activity[5–9].

Here, we focus on transcranial alternating current stimulation (tACS)[10], which has potential to affect neural oscillations relevant to normal cognition and neurological disease[11]. Specific oscillation frequencies characterize different arousal states and cognitive processes, and oscillations coordinate activity between local and distant brain regions[12, 13]. For example, in non-REM (NREM) sleep, neocortical slow-wave activity coordinates thalamocortical sleep spindles[14, 15]. It has been argued that low-frequency tACS, applied during NREM in healthy human subjects, can boost associative memory, presumably by entraining these nested rhythms[16–21]. Furthermore, low-frequency tACS applied during waking rest has resulted in widespread increases in theta activity, associated with improved encoding[22]. It has been proposed that tACS applied at the dominant network oscillations may entrain these endogenous rhythms in the gamma[23–25], beta[26], alpha[27, 28], theta[29], and slow frequencies[16, 30, 31].

In this study, we tested a widely cited protocol[16], which showed that declarative memory performance could be enhanced by applying low-frequency tACS during NREM sleep. This work has since inspired a number of replication studies, which have shown either positive results[18, 19, 21] or no effects[5, 7, 32, 33]. To validate the physiological underpinnings of this promising memory effect, we applied low-frequency tACS during sleep while measuring neural responses intracranially in surgical epilepsy patients. We test, in particular, the widely held belief that brain rhythms are entrained by tACS applied at the frequency that matches the endogenous rhythm. In the case of NREM sleep this is particularly advantageous as one can measure spindle and gamma activity, which is strongly coupled to the slow wave, yet is uncontaminated by the low-frequency stimulation artifact[34, 35].

Thirteen patients undergoing invasive monitoring for epilepsy surgery were tested with sinusoidal tACS (0.75 and 1 Hz) during a period of NREM sleep and/or waking rest. The primary outcome measure during sleep is phase-amplitude coupling (PAC), which captures the modulation of spindle amplitude (at 10 and 14 Hz) with the phase of tACS. This measure is motivated by the relevance of spindles to sleep-dependent cognitive processes, the strong entrainment of spindles to native slow oscillations[36], and the previous reports of tACS enhancement of endogenous slow-wave and spindle activity[16, 17, 20, 37, 38]. Our primary outcome measure during wakefulness is PAC of theta frequency (7 Hz) amplitude to tACS phase, as theta power has previously been implicated in tACS[22]. Secondary outcome measures include gamma (70–110 Hz) modulation during NREM sleep[39] and alpha (10 Hz) and gamma modulation during waking rest. We also tested on an additional subject the configuration of Marshall et al.[16] with identical electrode location, charge density, waveform, frequency, and intensity. Finally, we measured the intensity of induced electric fields at the cortical surface, and leveraged validated computational models to estimate field strengths across the brain[40]. We do not find an acute effect of the applied tACS on brain rhythms and attribute the outcome to the weak-induced fields.

As a positive control, we confirm that spindle power is modulated with the endogenous slow-wave rhythm during sleep in the same subjects across a majority of electrodes. Additionally, we find that acoustic stimulation with brief noise bursts reliably evokes slow-wave and related spindle activity comparable to effects found in healthy subjects using scalp electroencephalography (EEG)[41]. The null findings on entrainment together with these positive controls rule out the hypothesis that low-frequency tACS applied at conventional current intensities can acutely entrain slow-wave, spindle, gamma, or theta activity. We conclude that previous reports of behavioral effects of slow oscillating tACS applied during NREM sleep and waking rest on memory consolidation are not the result of direct neuronal entrainment. We discuss alternative mechanisms and propose new directions for research.

## Results

**Intracranial measurements of induced electric fields**. We applied low-frequency sinusoidal tACS (0.75, 1 Hz), on 13 patients with medication-refractory epilepsy with implanted

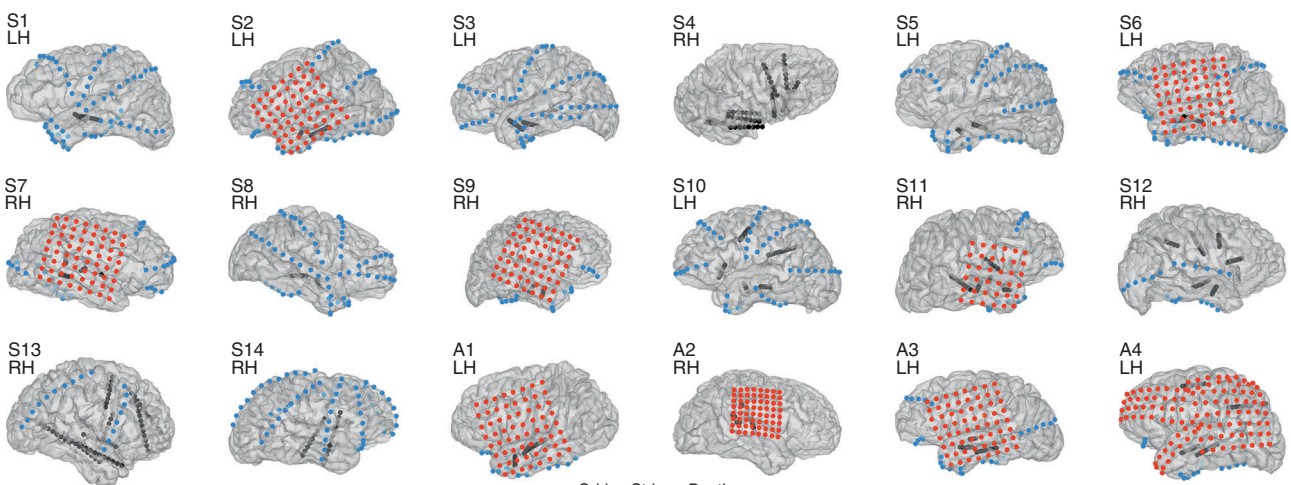

**Fig. 1** Electrode coverage for 18 patients who underwent invasive monitoring for epilepsy surgery and received either tACS or acoustic stimulation. Thirteen patients had tACS applied at a 1 or 0.75 Hz frequency, at stimulation intensities ranging from 0.5 to 2 mA, during waking rest (S1–S6) and daytime NREM sleep (S7–S13), one patient had trapezoidal tACS (S14), and four subjects received 0.75 and 1 Hz acoustic stimulation (A1–A4). Electrode placement varied by clinical indication, and consisted of a combination of strips, grids, and depth electrodes. Seven subjects had bilateral coverage (S1, S2, S3, S6, S8, S10, S13, S14). A total of 2156 electrodes total were utilized (1700 tACS; 113 Trapezoidal tACS; 343 acoustic), or an average of 120 electrodes per subject. Further demographic and clinical characteristics, electrode coverage, and stimulation protocols are summarized in Supplementary Table 1

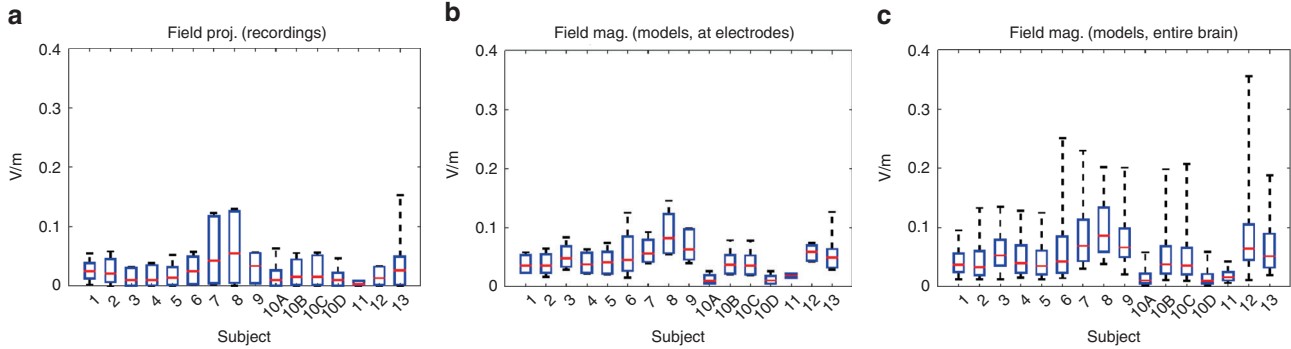

**Fig. 2** Measured and estimated electric field magnitudes. **a** Field projections calculated as the difference in recorded voltages between neighboring electrodes divided by electrode distance for each subject (with four montage orientations shown for S10), **b** field magnitudes at electrode locations predicted by calibrated current-flow models, and **c** model-predicted field magnitude across the entire brain. Red lines indicate the medians, and boxes span from 5 to 95% of the data, with whiskers extending to the minima and maxima. All values shown here correspond to the maximal current intensity applied for each subject during stimulation (S1: 1 mA; S2: 0.75 mA; S3: 1 mA; S4: 1 mA; S5: 1 mA; S6: 1 mA; S7: 1.5 mA; S8: 2 mA; S9: 1.5 mA; S10A: 0.3 mA; S10B: 1 mA; S10C: 1 mA; S10D: 0.3 mA; S11: 0.3 mA; S12: 1 mA; S13: 1 mA). The difference in magnitude across subjects is primarily due to these varying stimulation intensities

subdural and depth electrodes (Fig. 1; Methods; Supplementary Table 1). Overall, we recorded and analyzed signals from 1700 electrodes (mean 131 electrodes per subject) from the 13 patients receiving TES using this protocol. The positions of the recording electrodes sites in each patient are shown in Fig. 1. For stimulation, two scalp electrodes (2 cm × 2 cm rubber electrodes) were placed over the frontopolar and occipital poles, accessible under the surgical bandage. The aim was to generate the strongest possible fields in the brain by keeping electrodes at the furthest distance on the skull[42], while remaining apart from temporal craniotomies so as to replicate the current flow of normal head anatomy[43]. We confirmed that the craniotomy defect had only minor effects on field intensities as models of current flow with and without the defect yield comparable field intensities ($0.05 \pm 0.02$ V/m and $0.04 \pm 0.02$ V/m, respectively, mean ± Std across subjects). In one patient (S10), the stimulation electrodes were placed at three additional locations to examine the effects of electrode placement on extracerebral and intracerebral current spread. For three subjects (subjects S11–S13), frontal electrodes were offset to the left or right (Fp1 or Fp2) so that intracranial electrode coverage, which was largely constrained to one hemisphere, covered areas of maximal electric fields and thereby optimize recording of potential entrainment effects. Maximal stimulation intensity of tACS was adjusted for each subject based on amplifier saturation and patient sensation. Intensity ranged from 0.5 to 2.5 mA (current density 0.125–0.625 mA mA/cm$^2$), which is stronger than in previous reports[5, 7, 16, 18, 19, 32].

Since the electrical field, or the voltage gradient (V/m), induced in cortex critically determines neuronal polarization[44] and thus neuronal activity[4, 45], we measured the local gradients in electrodes across all subdural and depth electrodes. We calculated the electric fields in the direction of neighboring electrodes (i.e., the projected field; Fig. 2a) scaled to correspond to 1 mA of stimulation (voltages scale linearly with applied currents up to saturation levels of the amplifiers[40]). The median projected field measured across all electrode locations and subjects is 0.02 V/m. At the highest current intensity tested (2.5 mA), peak intensity reaches only 0.16 V/m across recording electrodes (S10).

To assess the induced electric fields over brain areas not sampled by electrode recordings and in the direction of maximal field intensity, we used previously established modeling techniques[40]. Briefly, computational models were individualized for each patient based on their magnetic resonance imaging (MRI) images and calibrated with the measured electric fields in each

patient. The models were then used to predict field intensities throughout the brain. For one patient (subject S13), electric field distribution for four different stimulation electrode configurations were estimated. The simulated electric fields predict the measured fields well ($r = 0.81 \pm 0.12$, mean and standard deviation across $N = 10$ subjects modeled, all $p < 10^{-5}$). The modeling results show larger fields for some locations of the brain as compared to the recorded fields (Fig. 2b, c), as expected given the limited electrode coverage. Maximal field magnitudes at hippocampal depth electrodes have a median of 0.05 mV/mm and a maximum of 0.11 V/m, for 2 mA stimulation intensity ($N = 8$ subjects). Median value across all the brains and electrode locations is 0.08 V/m (Fig. 2).

**Spindles entrain to the phase of endogenous slow waves.** Of the 13 subjects with tACS, 7 were stimulated with several 5 or 10-min blocks at 0.75 and 1 Hz during either daytime or nocturnal NREM sleep (Supplementary Table 1). Four subjects were stimulated during daytime NREM sleep; three were stimulated during nocturnal NREM sleep; and six were stimulated during waking rest. Duration and conditions of stimulation for each subject varied based on clinical constraints and the total time in NREM sleep (sleep stages N2 and N3, American Academy of Sleep Medicine 2012 convention[46]).

Sleep data from the seven subjects (subjects S7–S13) were analyzed over two separate nights without stimulation. We first visually identified periods of NREM sleep in the intracranial recordings. The average duration of the NREM recorded was $30.0 \pm 10.5$ min. Slow-wave activity spans broad frequency spectrum between 0.5 and 4 Hz with no clear peak in the frequency spectrum within these three octaves (Supplementary Fig. 6 and Supplementary Note 1) consistent with scalp EEG recordings in normal subjects[47–49]. Therefore, we analyzed activity and stimulated at both 0.75 and 1 Hz, values previously used in the literature[16, 22, 31]. As a primary outcome measure, we tested the modulation of spindle amplitude (for fast spindles: 14 Hz bandpass filter with 7 Hz bandwidth; for slow spindles: 10 Hz bandpass with 5 Hz bandwidth) by the phase of the endogenous slow oscillations (1 Hz bandpass filter with 1 Hz bandwidth). PAC was measured with the modulation index for individual electrodes and statistical significance was established using surrogate data with randomized phase (see "Methods"), here and in the remainder of the text. Recording sites that

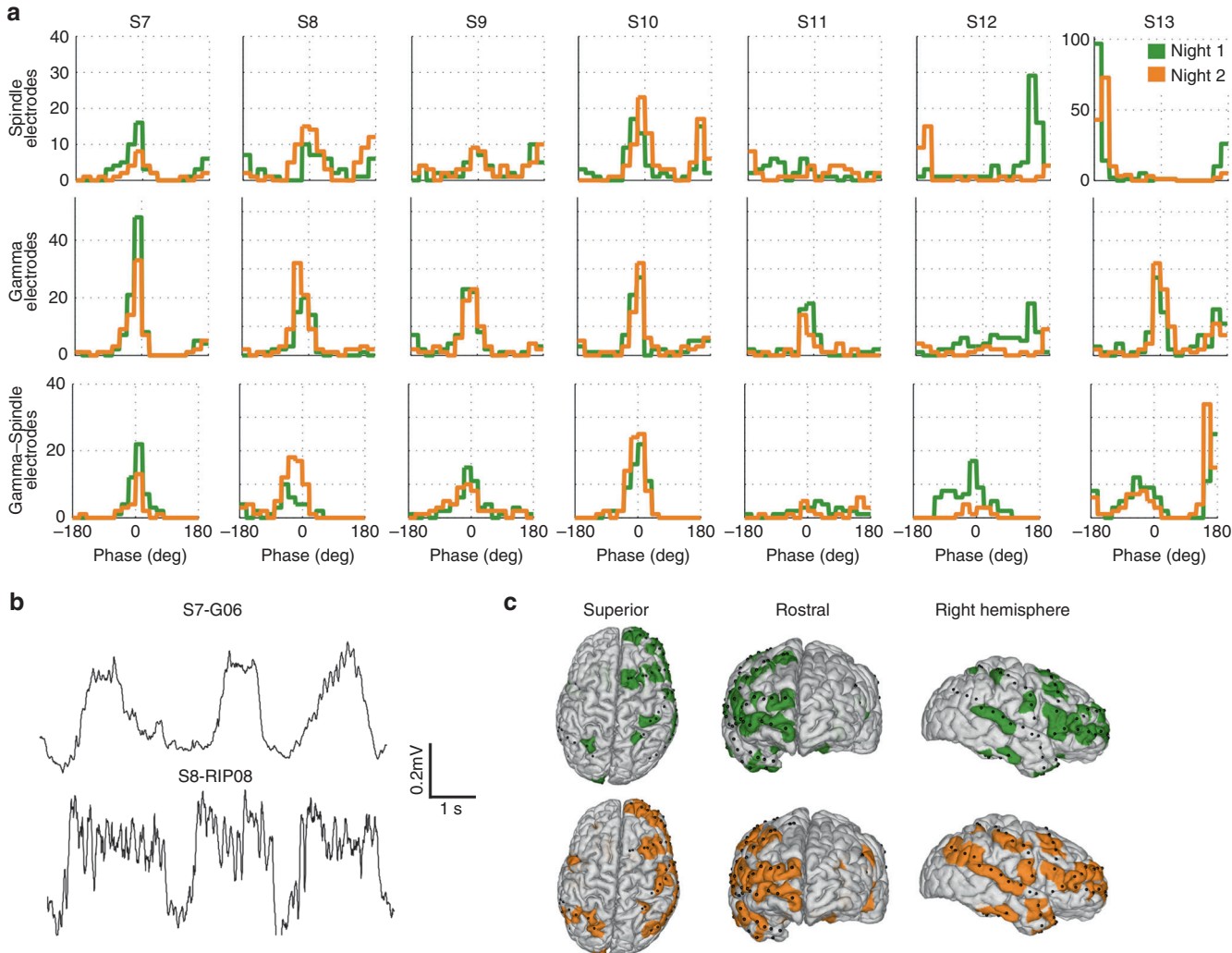

**Fig. 3** PAC of fast spindle and gamma activity to the phase of endogenous slow oscillation. **a** Histogram of the mean preferred phase (of 1 Hz bandpassed signal) across electrodes with significant PAC for two different nights (Night 1, green; Night 2, orange). Each histogram corresponds to a different subject. 0° corresponds to the positive phase of the raw iEEG traces, +/− 180° corresponds to the negative phase. The upstate is characterized by increased fast spindle (14 Hz) and increased gamma (70–110 Hz) activity. Electrodes with increased spindle/gamma activity in the positive phase (0°) are mostly cortical, and those with increase in the negative phase (180°) are mostly depth electrodes, which is consistent with previous reports[35, 49]. **b** Examples with spindle activity occurring with the positive phase of the slow-wave cycle as seen in the raw iEEG traces for two representative cortical electrodes (subject S7, electrode G06l; subject S8, electrodes RIP08). **c** Locations of cortical electrodes with significantly PAC of fast spindle activity for S8. Night 1 (top row, green) and night 2 (bottom row, orange) show widespread and consistent entrainment across nights. Left to right: top view, frontal view, right view. Black dots indicate the locations of the subdural grid electrodes

exhibited PAC are widespread over the cortex, representing most recorded regions (Fig. 3c). Of the 753 electrodes tested across subjects, 497 exhibit significant PAC (electrodes with $q < 0.05$ after FDR correction, electrode count summarized in Table 1 and Fig. 4b). Similar results are obtained with a 0.75 Hz bandpass filter (509 of 753 electrodes, $p = 0.55$, Fisher exact ratio test, Supplementary Note 2). Endogenous spindle power amplitude is highly modulated by the phase of endogenous slow waves (example electrodes with corresponding $p$-values in Fig. 4a). Seventy-seven percent of the channels showing PAC during night 1 also demonstrate PAC during night 2 (Figs. 3a, 4b, Supplementary Table 2, and Supplementary Fig. 6 for 0.75 Hz bandpass). The differences from one night to the other may be due to the differing depth of NREM sleep in the 30-min window tested, in addition to inter-individual differences.

We also computed the preferred phase of the spindle power within the endogenous slow-wave cycle for both nights and find

that it is stable across nights (angle difference averaged across electrodes: −9.7° ± 13.6° mean and Std across subjects, Fig. 3a, and Supplementary Fig. 7). 75 ± 16% (mean across all subjects) of the entrained electrodes have a preferred phase closer to 0°, but 25 ± 16% align closer to 180°. Slow waves reflect changes from cortical up and down states, with high and low neuronal firing activity, respectively[49]. For most electrodes, 0° corresponds to the upstate of the slow oscillation (Fig. 3b), associated with increased broadband gamma activity (70–110 Hz), which is a measure of neuronal firing[35] (Fig. 3a). Electrodes with 180° reflect a sign reversal of slow waves often observed in intracranial recordings[35]. As a secondary outcome measure, we computed the same PAC for gamma activity and found 533 of 753 electrodes entrained to the phase of the slow oscillation (Fig. 3a). When we calculated the difference of their preferred phase (Fig. 3a, Supplementary Table 2), we find that spindle and gamma activity cluster around a phase difference of 0° and coincide with the upstate of the slow oscillation.

**Table 1 Summary of PAC results across all subjects and conditions tested**

| Stimulation (subject ID) | Frequency | Arousal state | Subjects | Total # electr. | # electr. tested | Freq. band tested | # electr. entrained | Consistent across blocks (%) |
|---|---|---|---|---|---|---|---|---|
| None (S7–S13) | 1 Hz endogenous | NREM | 7 | 860 | 753 | 14 Hz (fast spindle) | 497 | 77 |
| None (S7–S13) | 1 Hz endogenous | NREM | 7 | 860 | 753 | 10 Hz (slow spindle) | 361 | 57 |
| tACS (S7–S13) | 1 Hz (.75 Hz) | NREM | 7 | 860 | 598 | 14 Hz (fast spindle) | 2 (1) | 0 |
| tACS (S7–S13) | 1 Hz (.75 Hz) | NREM | 7 | 860 | 598 | 10 Hz (slow spindle) | 1 | 0 |
| tACS (S1–S6) | 1 Hz | Wake | 5 | 716 | 584 | 7 Hz (theta) | 0 | 0 |
| tACS (S1–S6) | 1 Hz | Wake | 5 | 714 | 578 | 90 Hz (high gamma) | 1 | 0 |
| tACS (S1–S6) | 1 Hz | Wake | 6 | 840 | 695 | 10 Hz (alpha) | 8 | 0 |
| Acoustic (A1–A4) | 1 Hz (0.75 Hz) | NREM | 4 | 448 | 343 | 14 Hz (fast spindle) | 20 (51) | 4 |
| Acoustic (A1–A4) | 1 Hz (0.75 Hz) | NREM | 4 | 448 | 343 | 10 Hz (slow spindle) | 16 (33) | 4 |
| Trapezoidal (S14) | 0.75 Hz | NREM | 1 | 126 | 113 | 1 Hz (slow-wave) | 0 | 0 |

Columns indicate (from left to right): the type of stimulation, if any, along with the subject ID; the frequency of stimulation or analysis that provided the phase for the PAC measure; the arousal state of the subject; number of subjects tested; total number of electrodes tested across all subjects; frequency band that provided the amplitude for the PAC measure; total number of electrodes that showed significant entrainment in at least one block of stimulation/night; fraction of electrodes that entrained consistently across two or more blocks/nights of all electrodes tested

In addition to measuring PAC, we implemented a spindle detection algorithm[50] and computed their preferred phase within the slow-wave cycle. We obtain similar results with this spindle detection algorithm albeit with a smaller number of significantly entrained electrodes (187 instead of 497 of 753 electrodes; Supplementary Table 2).

**Low-frequency tACS does not entrain sleep spindles.** In four patients during an afternoon nap, and three patients during night-time sleep, we applied several 5–10-min blocks of continuous stimulation during NREM sleep (Stages 2 and 3). We tested tACS at 0.75 and 1 Hz consistent with previous studies[16, 17]. We reasoned that if the applied low-frequency tACS entrains endogenous slow-wave activity, then spindle power, which is coupled to slow-wave activity, should occur in phase with the applied fields. Because slow-wave activity cannot be directly assessed during tACS due to the stimulation artifact, spindle power (14, 10 Hz) was used as an index of the slow-wave rhythm. We used identical PAC processing and statistical tests as with endogenous sleep (modulation index $r$, assessed for significance with phase-randomized surrogate data), with the exception that phase could now be inferred exactly from the stimulation artifact, readily observed in the raw recordings (Supplementary Fig. 4). In subject S8, we find significant PAC of fast spindle power (14 Hz band) in one 1 of 116 electrodes after correcting for multiple comparisons across electrodes (FDR, $q < 0.05$; Fig. 4b, for this single electrode $N = 292$ cycles, $p = 0.0003$ prior for FDR correction). However, PAC in this electrode does not persist in following stimulation blocks separated by only 5-min intervals. The strength of evidence for PAC is compared across blocks in Supplementary Fig. 8. Other subjects (S7, S9, S10) do not demonstrate significant fast spindle entrainment (at $q < 0.05$) during any stimulation blocks. Figure 4b shows a summary of all electrodes tested at 1 Hz. For slow spindle power (10 Hz), only 1 of 98 electrodes demonstrates significant entrainment after correcting for multiple comparisons (FDR, $q < 0.05$, $r = 7.48$, $N = 216$ cycles, $p = 0.0005$ prior to FDR correction), but this was not maintained across stimulation blocks (Table 1).

To further increase the power of the statistical test we combined the data across tACS blocks. The total stimulation time tested after concatenating all blocks was 30 min (subject S7), 30 min (subject S8), 20 min (subject S9, 1 Hz), 10 min (subject S9, 0.75 Hz), 10 min (subject S10), 20 min (subject S11), 7.5 min (subject S12), 20 min (subject S13, 1 Hz), and 20 min (subject S13, 0.75 Hz). After concatenating all blocks and FDR correction, we find no electrodes demonstrating significant spindle entrainment at 14 or 10 Hz.

In three subjects (S9, S11, S13) we also tested entrainment of spindle power during tACS at 0.75 Hz as this frequency is a commonly used parameter in previously published studies[16, 17]. Figure 4b shows a summary of all electrodes tested at 0.75 and 1 Hz. Only one electrode (out of 93 total electrodes) in one of the three subjects (S9) tested demonstrates fast spindle entrainment with 0.75 Hz tACS comparisons (FDR, $q < 0.05$, $r = 1.18$, $N = 216$ cycles, $p = 0.0001$ prior to FRD correction), and only in one out of two stimulation blocks. In summary, no stable entrainment of spindle activity to the applied electric fields was detected for low-frequency tACS applied for intensities of up to 2.5 mA during NREM sleep.

**Low-frequency tACS does not alter spindle-gamma alignment.** Unlike applied stimulation waveforms, which remain steady during application, the endogenous slow waves remain periodic for at most two cycles (Supplementary Fig. 5), and are more often isolated events. This lack of temporal coherence is the source of the broadness of the slow-wave activity spectrum and lack of a well-defined peak in most electrodes (Supplementary Fig. 6). Because the endogenous slow-wave oscillation is not reliably discerned during stimulation due to the overwhelming stimulation artifact, we used gamma activity to infer the phase of the endogenous slow-wave rhythm. It is possible that tACS did not entrain spindle activity, but nonetheless interferes with the slow-wave rhythm, which is thought to coordinate activity in higher frequency bands. We therefore tested whether low-frequency tACS disrupts the temporal alignment of gamma and spindle activity, as measured by the cross-correlation of the instantaneous amplitude of these two rhythms (Fig. 5). As expected from the

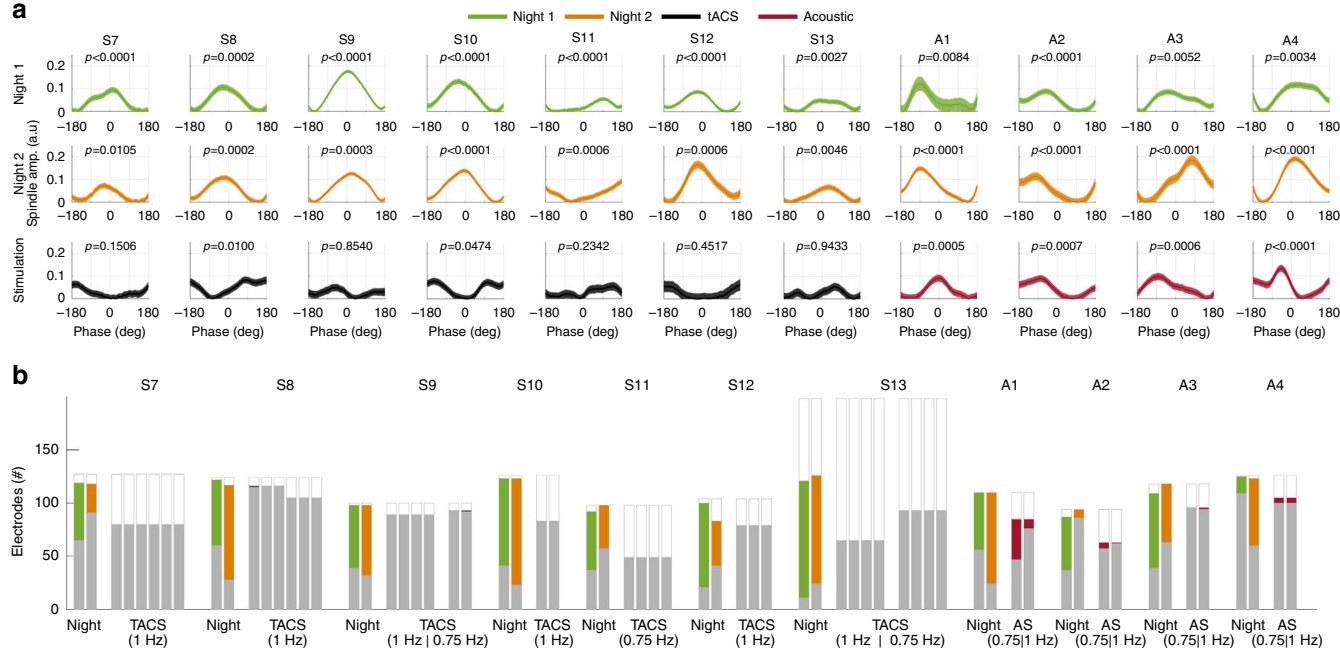

**Fig. 4** Modulation of fast spindle power with phase of the slow oscillation. Different conditions are indicated in color: endogenous sleep (green, orange), tACS (black), acoustic stimulation (AS, red). **a** Fast spindle power (14 Hz band) relative to the phase of slow oscillation shown in one representative electrode for each subject. Here, 0° phase corresponds to the physiological upstate of slow-wave activity. For endogenous slow-wave oscillation sleep periods (green and orange) phase is determined from 1 Hz bandpassed signal. For tACS (black traces) and acoustic stimulation (AS, red traces) blocks during nap (S7–S10, A2–A3) and during night-time sleep (S11–S13, A1, A4), phase is determined from the stimulation artifact or acoustic trigger pulses in the recordings, where 0° corresponds to the peak in anodal stimulation (e.g., peak positive stimulation relative to frontal/anodal electrode) for tACS, or the time of sound delivery in the case of AS. Note the modulation of spindle power with the phase of the endogenous slow oscillation in each subject during night-time sleep and during acoustic stimulation, and the lack of consistent modulation with the phase of tACS. Both nights have similar preferred phase. Each column represents the same electrode per subject. **b** Fraction of recordings sites with significantly entrained spindle activity during endogenous sleep, tACS, and acoustic stimulation. Each bar represents a block of data analyzed. Bar height indicates the number of electrodes available, with a fraction of electrodes discarded due to poor data quality or excessive interictal activity (white), a fraction of electrodes with non-significant entrainment (gray), and a fraction of electrodes with significant spindle entrainment (night 1, green; night 2, orange; tACS, black; AS, red) after correction for multiple comparisons. Each bar during tACS/AS represents a stimulation block of 5 min. During tACS stimulation only two recording sites (in S8, first block, red bar; and S09, second block of 0.75 Hz, red bar) showed significant spindle entrainment, although this was not a stable finding across stimulation blocks. With acoustic stimulation, many more electrodes were found to be reliably entrained across both 0.75 and 1 Hz stimulation rates

alignment of the two rhythms to the 0° phase or cortical upstate during endogenous sleep (Fig. 3a), the cross-correlation of the gamma and spindle power is maximal at zero-time delay (the median across electrodes and subjects S7–S13 $\pm$ the interquartile range is $0.0020 \pm 0.16$ s) for endogenous NREM sleep (Fig. 5, Nights 1 and 2). During stimulation, the temporal relationship of spindle with gamma activity (with a time lag of zero. Night 1: $-0.0156 \pm 0.020$ s; Night 2: $-0.0078 \pm 0.023$ s) is maintained (Fig. 5). This suggests that tACS does not disrupt the temporal coordination of spindle and gamma activity.

**Low-frequency trapezoidal tACS does not entrain slow waves.** To further test our null findings, we performed a close replication of the original study in which tACS was found to enhance slow-wave activity[16]. This was performed in one additional surgical patient (subject S14), during night-time sleep. We used the same montage, electrodes, waveform, and (lower) stimulation intensity of that earlier study[16] (bilateral ring electrodes on F3/F4 and mastoids with 0.75 Hz on/off trapezoid at 0.52 mA) and stimulated during NREM sleep periods. This electrode placement was facilitated in this one subject by a custom surgical bandage. The trapezoidal waveform does not permit analysis during stimulation due to the broad band artifacts they generate (for this reason we used sinusoidal stimulation for the majority of the experiments).

Thus, here we analyzed whether the slow-wave activity immediately after stimulation remained time-aligned across trials following previous methods[16, 37]. To achieve comparable statistical power to these earlier studies in a single subject we stimulated in 80 short blocks (8 cycles, or 10 s duration each; see also discussion on statistical power in Supplementary Note 3). After correcting for multiple comparisons (FDR, $q < 0.05$), none of the 113 electrodes exhibit phase-locked slow-wave oscillations (0.5–1.5 Hz) in the subsequent stimulation-free intervals (using Rayleigh test for non-uniformity, see "Methods"; uncorrected $p$-values reported in Supplementary Fig. 12). Remarkably, subsequent clinical evaluation did not reveal any electrical after-discharge or interictal abnormalities; the patient was diagnosed as non-epileptic.

**Low-frequency tACS does not modulate theta during wakefulness.** During waking rest, we also tested whether the applied field modulates theta activity (7 Hz bandpass, 3.5 Hz bandwidth) as a primary outcome measure, and alpha (10 Hz bandpass, 5 Hz bandwidth) or gamma activity (90 Hz bandpass, 40 Hz bandwidth) as exploratory outcome measures. We applied 1 Hz tACS to six patients during wakeful rest. After correcting for multiple comparisons (FDR, $q < 0.05$), none of the 584 electrodes tested for subjects S1–S4 and S6 exhibit significant PAC. Subject S5 was not analyzed for theta entrainment due to the presence of

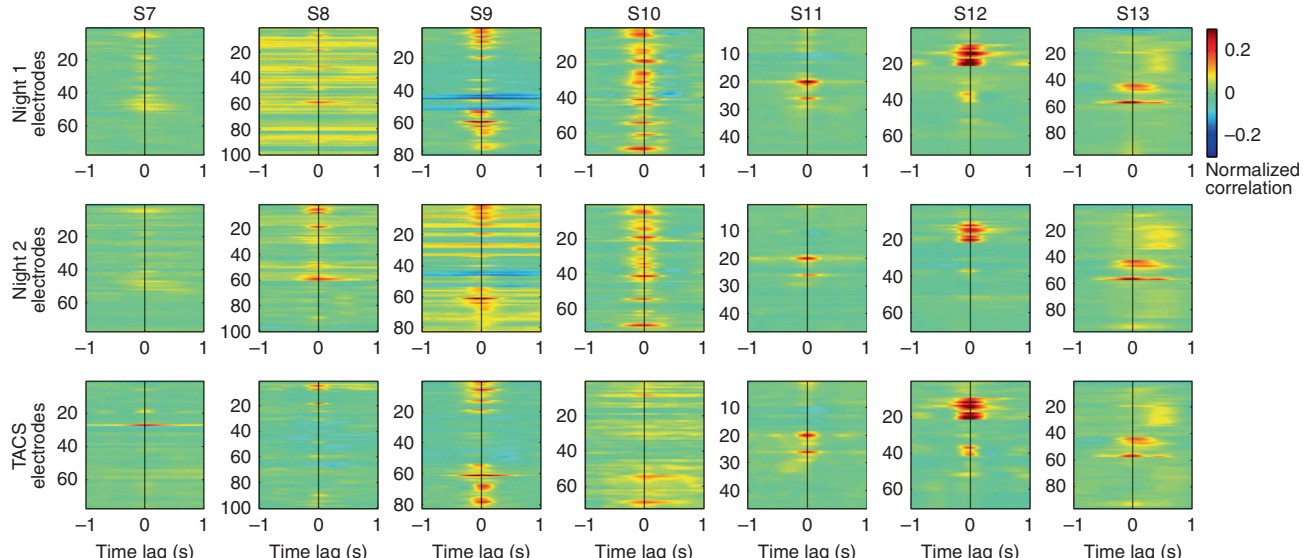

**Fig. 5** Cross-correlation between the amplitudes of spindle and gamma oscillations. Panel shows cross correlation for all electrodes for seven subjects (S7–S13) during two nights of sleep without stimulation and one with tACS. False color indicates correlation values. Vertical axis indicates electrode number and horizontal axis indicates time lag between the two rhythms. The peak at zero time-lag indicates that spindle and gamma occur at the same time, negative lag indicates that spindle activity precedes gamma activity

significant environmental artifact. For alpha activity, significant PAC is present in 1 of 122 electrodes tested in subject 1 (FDR, $q < 0.05$, $N = 594$ cycles, $p = 0.0004$ uncorrected), 2/78 electrodes in subject S4 (FDR, $q < 0.05$, $N = 594$ cycles, uncorrected $p$-values = 0.007, 0.003), and 5/155 electrodes in subject S6 (FDR, $q < 0.05$, $N = 594$ cycles, $p = 0.0004$, 0.0001, 0.0015, 0.0018, 0.0001, uncorrected). None of the 340 electrodes recorded for subjects S2, S3, and S5 demonstrate significant modulation of alpha activity with tACS phase. Significant coupling between stimulation phase and gamma activity is detected in 1/155 (FDR, $q < 0.05$, $N = 592$ cycles, $p = 0.0001$ uncorrected) recording sites for subject 6, but not in any of the 423 electrodes tested for subjects S1, S2, S4, and S5. Subject S3 was not analyzed for gamma power entrainment due to the presence of significant 60 Hz artifact. However, none of the recording sites with significant entrainment effects for theta, alpha, or gamma activity in one stimulation block show consistent effects across repeated stimulation blocks. We conclude that 1 Hz tACS does not entrain theta, alpha, or gamma activity during waking rest.

**Pulsed auditory stimulation induces slow-wave/spindle entrainment.** It could be argued that slow-wave activity cannot be reliably entrained in epilepsy patients. To directly address this possibility, we stimulated an additional four patients (subjects A1–A4) with low-frequency acoustic stimulation (50 ms, pink-noise bursts repeated at 0.75 and 1 Hz) delivered through earphones during nocturnal NREM sleep. Using the same analysis as described earlier (modulation index, assessed for significance with phase-randomized surrogate data), we find a number of electrodes that showed significant PAC entrainment during both 0.75 and 1 Hz auditory stimulation after FDR correction ($q < 0.05$; $p$-values prior to FDR correction are shown in Supplementary Fig. 10, Fig. 4; number of electrodes entrained for patient at 0.75 Hz, at 1 Hz, and consistently entrained across stimulation blocks as follows A1: 38, 11, 11; A2: 7, 1, 1; A3: 0, 2, 0; A4: 6, 6, 1). The lower numbers in A3 may be due to the lower sound volume used for sleep comfort. In total, we found 51/343 (14.8%) entrained electrodes during 0.75 Hz, and 20/343 (5.8%) entrained electrodes during 1 Hz acoustic stimulation, when pooled across all subjects. In addition, time-locked analysis of the individual electrode

waveforms relative to stimulation onset reveals robust slow-wave events and an increase in spindle power during the slow-wave upstate, when compared to the sham stimulation delivered during a baseline period of sleep ($\chi^2(99) = 200.23$, $p < 0.0001$, Fig. 6). These findings suggest that low-frequency auditory stimulation delivered during NREM can reliably entrain slow-spindle activity in epilepsy patients.

Finally, to exclude the possibility that we missed a genuine effect, we performed a number of additional post hoc analyses with less conservative thresholds, and limiting analysis to electrodes with highest field intensity, or those with marked spindle activity. Additionally, we tested for changes in spindle, slow-wave and theta power before and after stimulation. These analyses are detailed in Supplementary Note 4, and show no consistent effects.

## Discussion

We set out to test the hypothesis that low-frequency tACS can acutely modulate endogenous slow-wave rhythms during NREM sleep or theta activity during wakefulness. We find that low-frequency tACS, at 1 or 0.75 Hz, applied during NREM sleep at common stimulation intensities does not reliably entrain spindle oscillations during NREM sleep. This contrasts with spindle activity that is strongly entrained by endogenous slow oscillations in almost two-thirds of the electrodes in depth and cortical surface electrodes recorded during two nights of sleep. Gamma activity also reliably entrains to the upstate of slow-wave activity, at a similar phase as spindles. Furthermore, time alignment between the spindle and gamma rhythms is not altered by tACS, suggesting that co-occurrence of these rhythms to the cortical upstate was not disrupted by stimulation[51]. Despite medications, sleep physiology (i.e., coupling of spindle to slow activity during NREM sleep) in this patient population resembles that of healthy subjects[49]. In contrast, low-frequency auditory stimulation during NREM sleep evokes reliable slow-spindle events as seen in healthy subjects[41, 52]. Furthermore, low-frequency tACS during waking rest does not modulate theta, alpha, or gamma frequency activity with the phase of stimulation. Thus, low-frequency sinusoidal tACS does not entrain dominant rhythms during NREM sleep or waking rest.

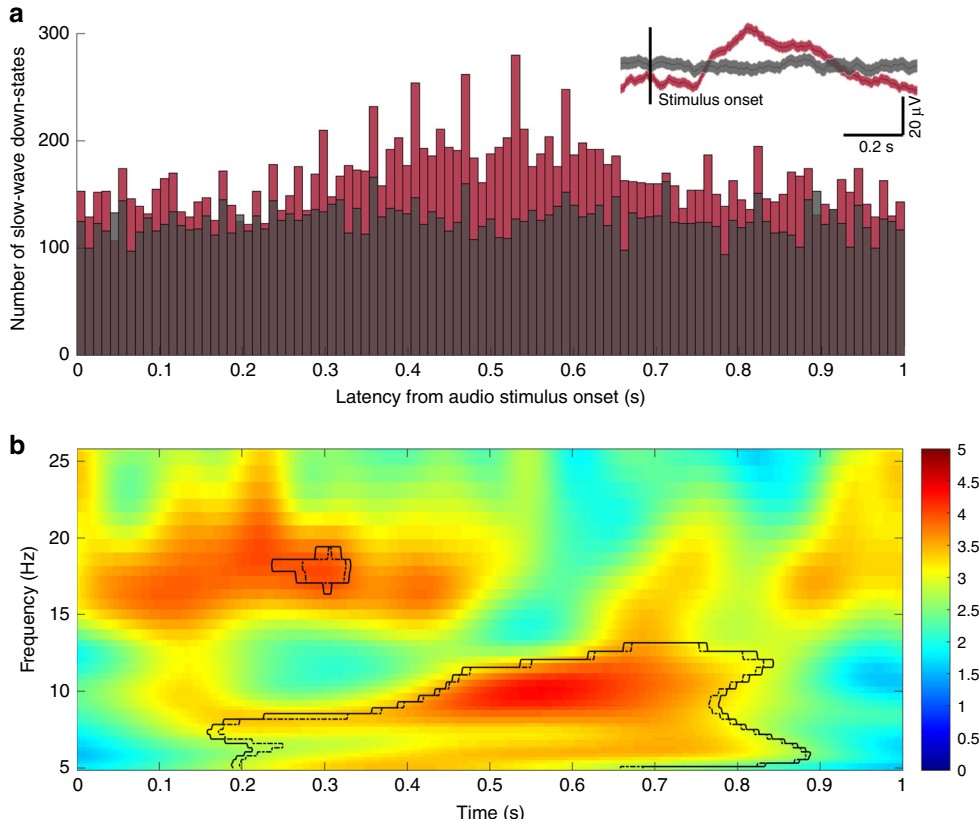

**Fig. 6** Slow-wave entrainment to 1 Hz acoustic stimulation. **a** Number of detected slow-wave oscillations relative to stimulus onset, summed over the 99 electrodes analyzed, in a 5-min interval for subject A4 during stimulation (red) and an equivalent baseline sleep period (black). Inset shows the averaged evoked response in both conditions in one representative electrode (depth electrode DPMT 03), time-locked to stimulus onset (note the positive peak a cortical down-state in this depth electrode, which is opposite from the majority of cortical electrodes shown as examples in Fig. 3b). **b** TFR of time-locked epochs (e.g., relative to stimulus onset) in one representative electrode (DPMT 03), showing an increase in slow spindle power (8–12 Hz) during slow-wave "down-states" consistent with previous literature[80] (0.4–0.8 s RE: stimulation onset) and fast spindle power (12–18 Hz) during "up-states" (0.15–0.4 s, RE: stimulation onset) as found for endogenous slow-wave oscillations (see Figs. 3, 4). TFR is computed relative to baseline sleep period with sham stimulation. Solid innermost curve represents significant increases relative to baseline after FDR correction (one-tailed $t$-test, $p < 0.05$)

It is tacitly assumed that tACS can entrain ongoing brain rhythms to boost memory performance[16, 18, 19]. However, previous studies have not directly measured entrainment of these latter rhythms. The present study is the first human experiment to assess the direct physiological effects of tACS by measuring cortical oscillations via invasive EEG. Here, we demonstrate that tACS does not necessarily induce coupled slow-spindle rhythms in human sleep. This specific result is important because of the exciting but mixed results reported with tACS and sleep-dependent memory enhancement[16, 18, 19], which highlights the lack of mechanistic understanding of this technique.

The strengths of our study depend on the simultaneous measurement of brain activity directly from the cortical surface during stimulation. Intracranial EEG (iEEG) provides superior signal quality compared to scalp EEG, allowing the direct measurement of induced electrical fields and their acute effect on mesoscopic brain activity. We collected a large data set with 18 subjects, 2156 electrodes (mean 120 electrodes per subject), and 900–3600 trials (oscillation cycles) per subject. Furthermore, to ensure that our null findings are not due to limitations in analytical technique, we included two positive control conditions, including endogenous sleep and acoustic stimulation, which demonstrate that slow-spindle rhythms can be entrained in these surgical patients and that the analytical methods can detect such entrainment.

Previous estimates of entrainment of slow-wave activity with applied slow tACS relied on averaging the periods immediately following stimulation, which gives a much smaller number of trials. In other words, measurement during stimulation with iEEG permits more sensitive detection of PAC as compared to measurement after stimulation with scalp EEG. Finally, we applied stronger stimulation intensities compared to previously published reports[16, 17]. In summary, our experimental and analytical approach permits detection of small changes in modulation, which we do not find.

Previously proposed explanations for how TES may affect neuronal activity rely mainly on in vitro slice preparations and in vivo experiments in rodents[53]. In vitro studies demonstrate that a weak uniform (DC) electric field can alter the resting transmembrane potential, which increases or decreases average firing rates of the affected neurons[23, 44]. In vitro studies using AC stimulation demonstrate single neuron entrainment by modulating the rate and timing of neural firing[23, 30, 54]. It is assumed that small changes in the firing rate and coherence of the affected neuronal populations may be amplified by network-level activity[23, 54]. However, the weak effects of the applied field also compete with the endogenous neuronal drive, which controls the instantaneous phase and amplitude of the rhythmic activity. For stimulation to entrain neural activity that does not have a steady rhythmic behavior, such as slow-wave activity, one would need fairly strong effects that reset phase in a single cycle. This may explain why several-fold stronger currents are needed to entrain some network oscillations compared to the lower intensities required to alter firing rates of several neurons[30, 31].

Our null findings likely result from the weak-induced electrical fields. At the highest current intensity applied (2.5 mA), peak intensities reach only 0.16 V/m across recording electrodes. Although the recording electrodes may not capture the maximal electric fields induced by a given stimulation montage, our model-based extrapolations suggest that peak fields do not exceed 0.43 V/m intracranially when stimulating at 2 mA[40], which is comparable to a recent report of 0.5 V/m as maximal intensities[55]. Thus, field magnitudes may be an order of magnitude smaller than required to entrain slow-wave rhythms, which may explain why experimental attempts to entrain sleep rhythms for memory enhancement have been inconsistent[5, 7]. To place these weak-induced fields into perspective, synchronous neuronal activity under physiological conditions can generate approximately 1–2 V/m gradient across the CA1 pyramidal layer during theta oscillations in the hippocampus[56] and during slow oscillations in the neocortex[30]. Under ideal circumstances when the electric fields are applied in a phase-locked manner coupling directly with the cellular membrane in vitro, 0.25 V/m can bias a single neuron's spike threshold to induce cumulative effects and entrain network oscillations[23, 57]. In behaving rats, hippocampal firing rate, but not local field potential oscillations, was modulated at a threshold of 1 V/m[31]. In other scenarios, 5 V/m or more may be needed to entrain network oscillations[30, 53].

The weak electric fields measured in our study suggest that higher intensity stimulation may be required to instantaneously affect network-level neuronal activity. The practical implication of our findings is that new methods of neuromodulation, which increase the amount of current reaching the brain, are needed to advance the field. However, transcranial stimulation intensity is limited by clinical considerations such as sensory perception. These constraints are especially relevant for sleep studies (one of our subjects was wakened by >2 mA stimulation and we had to decrease the intensity), subject blinding[58], and patient safety[59].

Another possible explanation for our null findings is the observation that endogenous slow-wave activity does not have a steady rhythm, but rather spans wide frequency range (0.5–4 Hz). In human sleep, slow waves appear to be unitary events, and a regular rhythm is preserved for at most two cycles. Thus, the continuous application of a weak, low-frequency stimulation waveform aligns imprecisely to these irregular events. In order for spindle activity to align to the stimulation phase, the effect of stimulation would have to be nearly instantaneous. This would require pacing of the endogenous rhythm on a nearly cycle-by-cycle basis. By contrast, the most sensitive effects of stimulation[23] (at weak fields of 0.2 V/m) occur when the endogenous rhythm is narrow-band such that stimulation effects can accumulate over many cycles, leading to resonant phenomena. Our results therefore do not rule out more sensitive resonant effects of tACS for more rhythmic narrow-band activity such as alpha and spindle oscillations[38].

Finally, these null effects may result from the anatomical differences seen when scaling from in vitro to rodent to human studies. Due to complex cortical folding in humans, portions of the network that are differently oriented relative to the induced potential field may demonstrate both excitatory and inhibitory behavior.

We acknowledge the difference in stimulating electrode placement between our protocol and previously published protocols, yet do not conclude that the null effect is because of this experimental difference. Because of the clinical limitations of the surgical bandage, we were able to place electrodes in the frontopolar and occipital regions, directly below the surgical dressing. The advantage of maximizing electrode distance as in our protocol is to decrease the amount of current shunted by the scalp (compared to electrode placement in the bilateral frontal regions

(F3/F4) and mastoids as prior). This generally should lead to stronger electric fields in the brain as compared to the F3/F4 montage[42], as supported by our recently validated current flow models[40].

The positive control of acoustic stimulation demonstrates that slow-wave and spindle activity can be modulated during NREM sleep in our patients. Acoustic stimulation has previously been shown to enhance sleep oscillations in healthy subjects[52] with an increase in sleep-dependent memory consolidation[41]. Using a low-frequency acoustic analog of tACS during NREM, we observe sleep spindles are modulated by the rhythmic stimulation in 8–14.3% of the electrodes analyzed. Additionally, the timing of spindle activity coincides with the upstate for both acoustic stimulation and endogenous slow waves. Together this suggests that endogenous slow wave may have been entrained by the rhythmic stimulation. Regardless of the mechanism (entrained vs. induced spindle), this indicates that modulation of sleep oscillations is possible, and that the analytic techniques used to assess entrainment are sensitive to the PAC of slow-wave/spindle events.

There are, however, limitations of our study. First, because the recording electrodes are placed according to clinical indication, they do not necessarily capture the maximal electric fields induced by a given stimulation montage. Despite broad coverage of the cortical surface, peak electric fields may occur outside the recorded areas (but should not exceed 0.43 V/m for 2 mA). Nonetheless, spindle entrainment during endogenous sleep and acoustic stimulation show the widespread nature of these cortical rhythms. Even if our electrodes did not capture the maximal electric fields, we would have expected to see a small number of electrodes entrained from tACS. Second, while we did examine changes in power before and after stimulation, the experiments were not optimally designed for this purpose, and thus fluctuations that were found cannot be attributed to tACS, but are likely the result for natural fluctuations of power within and across nights of sleep. Third, we only considered low-frequency stimulation. Given the dependence of oscillatory stimulation of the specific of network rhythms[53], these findings may not generalize to other tACS frequencies. In summary, as with all null results, one cannot rule out effects outside the parameters tested here. In particular, it is possible that there were lasting effects on spindle power, which we did not resolve here because of their natural fluctuations during sleep on the time scales of minutes. Such changes were reported in past studies[16, 21, 60] by averaging over many subjects and may have resulted from the net-DC currents used there, which we did not test here.

Patients with refractory focal-onset epilepsy undergoing surgical evaluation represent a unique opportunity to record directly from the cortical surface during non-invasive stimulation. While aberrant local networks involved in seizure propagation are observed, other cortical and subcortical functions are often considered normal. The potential effects of antiepileptic medications may decrease overall excitability, although some patients were tapered off all medications during stimulation. Widespread and strong spindle entrainment by native slow oscillations during sleep demonstrates that our methods detected physiologically meaningful changes despite medication use, similar to what has been reported previously regarding sleep rhythms recorded in epilepsy surgical patients[61]. We demonstrate that entrainment of spindle activity in epilepsy patients is possible with acoustic stimulation. Finally, an exact replication experiment of previously published protocols, delivered to a healthy brain, confirmed the absence of acute entrainment of slow waves and further strengthens the generalizability of our results.

We emphasize that our null results for tACS do not contradict the reported behavioral effects. While positive behavioral results have been found in rodents[20], a meta-analysis on memory effects

reports mixed results in humans[33]. There may be other means by which non-invasive stimulation affects brain activity in multiple indirect ways, including activation of afferent nerves[58, 62], retina and the vestibular apparatus[63], astrocytes and perivascular elements[64, 65], glial activation[65], synaptic plasticity[66], as well as through placebo effects[67], which merit further exploration.

Our findings suggest that investigation of novel methods of stimulation delivery are needed. This may include methods that induce stronger electric fields at the cortical surface (while minding patient safety and sensation), or may be based on acoustic and other modes of sensory stimulation. Furthermore, applying current at the optimal phase of endogenous rhythms in a closed-loop manner may be more effective[68]. However, such responsive methods can be therapeutic only if the impaired network pattern is identified and continually monitored[69, 70].

In some medically justified cases, chronic closed-loop feedback stimulation may be beneficial. Electrode plates may be placed directly onto the skull or on the cortical surface to bypass the skin, as illustrated by an FDA-approved therapeutic device, used to detect and disrupt electrographic seizures[71]. Further progress requires the investigation of novel electrode arrangements and stimulation delivery to produce meaningful and reproducible physiological and behavioral effects.

At a minimum, the present data suggests that tACS at stimulation intensities of up to 2.5 mA does not entrain slow-wave activity during NREM sleep. More generally, our result challenges the common assumption that tACS entrains and enhance endogenous rhythms. Thus, future studies will now have the burden of proof when such claims are made.

## Methods
**Human subjects**. This study was performed in epilepsy patients undergoing surgical evaluation with iEEG monitoring at New York University Medical Center (NYUMC). The protocol was approved by the NYUMC Institutional Review Board and the Clinical Trials Registration number was NCT02263274 (www.clinicaltrials.gov). Subjects were eligible according to pre-established criteria, including: (1) age over 18 years old; (2) undergoing invasive monitoring for seizure localization for epilepsy surgery; and (3) ability to provide informed consent or have a legal guardian who could consent. Exclusion criteria included (1) significant cognitive impairment (IQ < 70), (2) facial or forehead skin breakdown that would interfere with surface electrode placement, (3) contraindication to MRI, (4) known adhesive allergy, (5) space occupying lesion, and (6) subjects who had an electrographic seizure for 1 h prior to the stimulation procedure. All patients ($n = 17$) or their caregivers provided informed consent. Subjects were enrolled between December 2013 and May 2017. A table listing subject characteristics is included in Supplementary Table 1.

**Sleep staging**. Stimulation was performed after patients had entered at least 5 min of continuous NREM sleep, during a daytime nap or nocturnal sleep. Initial sleep staging was performed by visual online analysis, for the presence of a slow-wave and spindle activity as detected in the real-time iEEG seen at the bedside by a physician board-certified in clinical neurophysiology, as well as by direct clinical observation. As part of standard practice at NYU, an extradural lead is customarily placed near the vertex of the craniotomy to aid with spindle detection. When the patient aroused or drifted into a lighter stage of sleep, stimulation was stopped. The iEEG segments were later confirmed by a second board-certified physician to be consistent with N2 and N3 sleep. During offline processing, we selected segments of NREM sleep for analysis by comparing raw spectrograms of sleep (after artifact subtraction), to demonstrate that the depth of NREM sleep is similar across testing conditions (Supplementary Fig. 6).

**iEEG recordings**. iEEG was recorded from implanted subdural platinum-iridium electrodes embedded in silastic sheets (2.3 mm diameter contacts, 10 mm center–center spacing, Ad-Tech Medical Instrument, Racine, WI) or depth electrodes (1.1 mm diameter, 5–10 mm center–center spacing). The decision to implant, placement or recording electrodes, and the duration of invasive monitoring were determined solely on clinical grounds and without reference to this study. Electrodes were arranged as grid arrays (8 × 8 contacts, 10 or 5 mm center-to-center spacing), linear strips (1 × 4 to 12 contacts), or depth electrodes (1 × 8 or 12 contacts), or some combination thereof. Subdural electrodes covered extensive portions of lateral and medial frontal, parietal, occipital, and temporal cortex of the left and/or right hemisphere.

Within 24 h after surgical implantation of electrodes, patients underwent a post-operative brain MRI to confirm subdural electrode placement. Electrode were localized and mapped onto the pre-implant and post-implant MRI using geometric models of the electrode strips/girds and the cortical surface[72].

Here, we present an efficient method to accurately localize intracranial electrode arrays based on pre-implantation and post-implantation MR images that incorporates array geometry and the individual's cortical surface.

**Clinical (macroelectrode) recording equipment**. Recordings from grid, strip, and depth electrode arrays were made using a NicoletOne C64 clinical amplifier (Natus Neurologics, Middleton, WI), bandpass filtered from 0.16–250 Hz and digitized at 512 Hz. ECoG signals were referenced to a two-contact subdural strip facing toward the skull near the craniotomy site. A similar two-contact strip screwed to the skull was used for the instrument ground.

**NeuroConn DC stimulator**. The DC-STIMULATOR PLUS (NeuroConn, Germany) is a CE-certified medical device for conducting noninvasive TES in humans. The stimulator is a micro-processor-controlled constant current source, which continuously monitors electrode impedance, and detect insufficient contact with the skin. The device is battery powered, and therefore electrically isolated from the clinical recording electrodes and equipment.

**Low-frequency tACS**. We performed 0.75 and 1 Hz sinusoidal tACS on 13 epileptic patients with implanted subdural and depth electrodes. Seven subjects were stimulated during NREM sleep (four daytime nap; three nocturnal sleep) and six subjects were stimulated during waking rest, eyes closed. Patients were over 18 years old and fluent in English. Subjects were excluded if they had frequent (>2) electroclinical seizures in the 24 h preceding stimulation. Patient characteristics and electrode coverage are summarized in Supplementary Table 1 and Fig. 1.

All subjects tolerated scalp stimulation. All subjects who were stimulated during night-time sleep ($N = 3$, subjects 11–13) and most subjects during an afternoon nap ($N = 4$, subjects 7–10) were able to be sleep through trials at stimulation intensities between 0.5 and 2 mA. One subject (subject 13) woke from sleep and reported an itching sensation during one stimulation block with 2.5 mA current intensity. For the tACS experiments, we recorded and analyzed from 1700 electrodes without artifacts (mean of 131 electrodes per subject, example electrode in Fig. 3b). There were no complications from stimulation, and no induced electrographic seizures. One patient (subject 7) had a typical electroclinical seizure during stimulation. Because this patient had frequent spontaneous seizures, it was determined by the patient's epileptologist that stimulation was unlikely to have caused the seizure.

Furthermore, we enrolled one subject (S14) who had a bilateral subdural strip and depth survey, to perform a precise replication experiment of prior protocols. This patient had multiple target clinical events captured, which were non-epileptic in nature. He did not have any interictal or ictal activity captured during 1 week of monitoring, even while medications were being withdrawn. In other words, the patient did not demonstrate any epilepsy-related pathophysiology.

We reviewed the hour of iEEG recording prior to stimulation to exclude recent seizures. We performed a pre-stimulation clinical assessment (including assessment of the stimulation skin site and neurologic examination). A physician (AL) was present at the bedside during the entire procedure to monitor for safety. The patient's iEEG recording was monitored in real time at the bedside during stimulation for seizures.

For patients S1–S13, two stimulating electrodes were placed medially over the frontal and occipital poles (2 cm × 2 cm rubber electrodes) for patients S1–S10. In patients S10–S13 the frontal electrode was offset from midline by 3 cm (S10 and S13 left frontal; S11 and S12 right frontal) to minimize distance from stimulating electrodes to recording electrodes. In one patient S10, the stimulation electrodes were placed at three stimulation locations to examine the effects of electrode placement on extracerebral and intracerebral current spread. Subjects were covered with a nickel-cadmium shroud to reduce environmental artifact during recording, and other sources of environmental noise (60 Hz) were minimized in the patient area. The stimulation protocol used the NeuroConn DC Stimulator Plus (NeuroConn, Germany), with a biphasic sinusoidal waveform at 0.75 and 1 Hz, at variable intensities between 0.3 and 2 mA, for 10 s (cycles) to determine the peak intensity at which amplifier saturation occurred. Thereafter, subjects were stimulated with TES at 0.75 and 1 Hz, at variable intensities up to the peak intensity, for a duration between 5 and 10 min (10 min for A1–A6 and 5 min for S1–S4). Up to four blocks of stimulation were applied, until subjects woke up. The more than ten-fold increase of the subdurally recorded iEEG amplitude, compared to the EEG signal[73], allowed for simultaneous recording and stimulation (up to saturation levels of the amplifiers). Stimulation was immediately stopped in the event of an electrographic seizure (S7). A repeat clinical assessment (including assessment of stimulation skin site and neurologic examination) was performed after stimulation.

For S14, who was enrolled to perform a replication experiment of prior protocols, we selected a surgical patient who had a bilateral strip and depth survey. There were two windows that were cut into the patient's surgical bandage to allow electrode placement at the F3/F4 positions. Stimulation electrodes (8 mm Ag/Cl ring type) were applied bilaterally, with anodes at F3/F4 and cathodes on each mastoid. To test for acute effects, we utilized a stimulation protocol using

trapezoidal waveform (0.33 s ramp up/0.33 s steady state/0.33 s ramp down/0.33 s zero current), 0.75 Hz, 0–0.26 mA, for 8 cycles on/8 cycles off (10.66 s ON/10.66 s OFF), for 80 cycles. Stimulation was started after the first 5 min of NREM sleep.

**Low-frequency noise-burst auditory stimulation**. Auditory stimulation consisted of 50 ms pink-noise bursts (1/f spectrum) repeated at a rate of 1 of 0.75 Hz to four patients (subjects A1–A4). Sound was delivered via flat-profile headphones (Bedphones, Millwood, NY), which were placed on the patient's ears. Placement of the headphones required access under the surgical bandage and was administered by an epilepsy physician (AL), with patient verification of correct positioning. Acoustic pulses (50 ms, pink noise, 5 ms on/off ramps) were digitally generated and delivered via a laptop placed at the bedside. The sound level of the stimulation was manually adjusted for each patient to maximize comfort (e.g., ability to sleep with sound playing in the background). The volume was recorded and estimates of the sound level presented to each subject were assessed via an ear simulator (KEMAR Head and Torso simulator, Knowles Research, coupled to a B&K type 3134 Pressure microphone and B&K type 2230 Sound Level Meter, Bruel & Kjaer, Denmark). Resultant peak sound level estimates for each subject are 72 dB SPL (subject A1), 68 dB SPL (subject A2), 46 dB SPL (subject A3), 70 dB SPL (subject A4).

Similar to the procedure used during tACS stimulation, acoustic stimulation was presented at repetition rates of 0.75 and 1 Hz, in blocks of 5 min during NREM sleep. For each subject, we collected a block at each stimulation rate during NREM sleep, which was visually confirmed offline. The order of the presentation blocks was randomized across subjects. An awake control condition was performed for each patient, to verify the presence of acoustic ERPs. Additional time-locked TTL triggers were generated for each stimulus presentation and recorded by the EEG amplifier's DC input to aid in offline analysis.

**iEEG data preprocessing**. All electrodes were inspected for signal quality by plotting spectrogram, raw voltage, and the power spectrum. We recorded from a total of 1700 electrodes for this analysis. Electrodes were discarded based on high 60 Hz noise (likely due to poor contact impedance), amplifier saturation (clipping), or poor removal of the tACS artifact (due to non-stationarity of stimulation artefact, typically resulting from patient movement). Example of artefact-free recording electrodes are shown in Supplementary Figs. 2 and 3. The electrodes remaining for each subject during tACS were: 122/126 (S1), 112/126 (S2), 117/126 (S3), 78/84 (S4), 111/124 (S5), 155/254 (S6), 80/128 (S7), 116/122 (S8), 89/100 (S9, 1 Hz tACS), 93/100 (S9, 0.75 Hz), 83/124 (S10), 49/98 (S11), 79/102 (S12), 88/188 (S13, 1 Hz tACS), 93/188 (S13, 0.75 Hz tACS). During endogenous sleep the electrodes remaining for each subject during night 1 were: 123/126 (S7), 103/122 (S8), 96/100 (S9), 91/124 (S10), 70/98 (S11), 94/102 (S12), 170/188 (S13). During night 2: 121/126 (S7), 113/122 (S8), 89/100 (S9), 96/124 (S10), 82/98 (S11), 93/102 (S12), 164/188 (S13).

**Measurement and modeling of electric fields**. During tACS, the current alternates in directionality between two stimulating electrodes. This alternation results in a sinusoidal signal that can be used to determine the magnitude of the stimulation voltages. Magnitude was estimated by fitting a sinusoid to the signal at each electrode location and estimating the magnitude of the fitted signal. The output of this processing was plotted and manually inspected electrode by electrode.

The measured voltage in each location is then used to derive the projected electric field, by subtracting potential values between adjacent electrode pairs and dividing by their distance, resulting in V/m. The adjacent electrode was defined as the closest electrode within a 10 mm vicinity for cortical electrodes on the same grid array and linear strip, and 5 mm vicinity for depth electrodes on the same strip, to reflect the different inter-electrode distance. It is important to realize that this only captures a fraction of the field magnitude at any given location as the field orientation may not be parallel to the direction of two neighboring electrodes. The distant stimulating electrodes were expected to generate the strongest field intensities on the cortical surface directly under the scalp electrodes[40]. However, recording electrodes lay predominantly orthogonal to field direction (parallel to the cortical surface). Thus, the measured field projections will not capture maximal intensities, except in the rare circumstance that a depth array is precisely underneath one of the stimulating electrode and oriented toward a second, distant stimulating electrode.

The computational models were built following our previous work[74]. Briefly, the MRI for each patient was automatically segmented by the New Segment toolbox[75] in Statistical Parametric Mapping 8 (Wellcome Trust Centre for Neuroimaging, London, UK) in Matlab (R2013a, MathWorks, Natick, MA). Segmentation errors were corrected first by a customized Matlab script[74] and then by hand in ScanIP software (v4.2, Simpleware Ltd., Exeter, UK). The field of view of the clinical MRI scans was extended down to the neck by co-registering a standard head[74], and pasting the lower portion of the standard head to the model. The 2 × 2 cm stimulation electrodes were positioned on the model using CAD software. For each patient, a finite element model was generated from the segmentation data and then the electric potential distribution was computed assuming 1 mA current through the stimulation electrodes. Tissue conductivities were adjusted to minimize the mean-square difference between predicted and measured potentials. With these calibrated models, we then computed electric

fields throughout the brain. Electric potentials of model and measurements corresponded closely, with correlation values of $r = 0.95 \pm 0.04$ (mean ± Std across patients, $N = 1545$ electrodes across 10 subjects). Electric field is the spatial derivative of these potentials. They are estimated as the difference in electric potential between neighboring electrodes, divided by the distance. This is the electric field projected on the orientation of the electrode pair[40].

**Phase-amplitude coupling**. PAC measures the degree to which the amplitude of a high-frequency oscillations, $A_{HF}(t)$, is aligned with the phase of a lower frequency, $\phi_{HF}(t)$. We were interested in the interaction between the amplitude of spindle activity band at 14 Hz with the phase of endogenous slow oscillations at 1 Hz (or 0.75 Hz) during sleep as well as entrainment to the applied stimulation (tACS and acoustic). This section refers to 14 Hz activity, but the identical analysis was done for power amplitudes in the theta, alpha, and gamma bands (see next paragraph). To measure entrainment of spindle oscillations during endogenous sleep, we compared spindle power against the phase of the endogenous slow oscillation activity ($\phi_{LFE}(t)$, low-frequency endogenous). To measure entrainment to tACS, we used the phase of the electrical stimulation artifact ($\phi_{LFS}(t)$, low-frequency stimulation; Supplementary Fig. 1). To obtain the phase during acoustic stimulation we used the delay from the onset of each noise burst. To obtain the phase during tACS and remove the stimulation artifact, we first modeled the 1 Hz artifact as a linear superposition of sines and cosines at multiples of a base frequency (harmonics up to 40 Hz) by fitting the amplitude of each sine/cosine and the base frequency. An example of this fitting procedure is shown in Supplementary Fig. 4 with the top row indicating the raw signals and the bottom row showing the signals after the fitted harmonic artifact has been subtracted. The resulting harmonic fit captures the 1 Hz stimulation artifact including any harmonic distortion that may have resulted from amplifier nonlinearities. We calculated the stimulation phase from the harmonic fit (Supplementary Fig. 1D). To obtain the phase during endogenous sleep, first we applied a complex-valued Morlet wavelet filter centered at 1 Hz (or 0.75 Hz) with a bandwidth of 1 Hz (or 0.75 Hz; in humans, the center frequency of slow-wave activity is often assumed to be <1 Hz)[48]. The instantaneous phase can be directly extracted from the complex-valued filtered signal. Peak and trough of the slow oscillation are indicated by 0° and 180°, respectively, which represent the cortical upstate and downstate as discussed in the main text.

To obtain the instantaneous amplitude of the high-frequency rhythm during tACS we subtract the harmonic fit (Supplementary Fig. 4) and filtered the residual iEEG signal with a complex-valued Morlet bandpass for spindle activity ($fc = 14$ Hz, bandwidth = 7 Hz), for alpha activity ($fc = 10$ Hz, 5 Hz bandwidth) and for gamma activity ($fc = 90$ Hz, bandwidth = 40 Hz, Supplementary Fig. 1C). The instantaneous HF power was obtained by taking the absolute value of the filtered signal (Supplementary Fig. 1E). Patient movement and other artifacts resulted in outliers in the HF amplitude estimates. We removed these by removing cycles with excessive power as follows. For each electrode and each LF cycle we compute the mean power of the HF band (square amplitude averaged over one LF cycle). Cycles are removed as outliers if their mean power exceeds two times the interquartile range across all cycles in an electrode. To obtain the amplitude of the high-frequency rhythm during endogenous sleep and auditory stimulation we used the same procedures starting from the raw iEEG signal (no harmonic fit is needed). Outlier rejection was done as before based on the mean power in the high-frequency band. In addition, for endogenous slow wave we excluded cycles for which the slow oscillation amplitude was below 50 μV. For the analysis of the acoustic stimulation we used the same procedures as with the endogenous sleep except that the phase was defined based on the time since the onset of the noise burst, and cycle duration determined from the average inter-stimulus interval (TTL pulse) for each subject.

PAC is measured here using the modulation index $r$, which is defined as the absolute value of the time average, $r = |<z(t)>|$, of the complex-valued quantity, $z(t) = A_{HF}(t)e^{i\phi_{LF}(t)}$ (Supplementary Fig. 1F). Time average $<z(t)>$ is computed over cycles and time within a cycle. If $z(t)$ does not have a radially symmetric distribution this will cause the time average $r$ (modulation index) to be different from zero. This can be a result of (1) the amplitude $A_{HF}(t)$ is consistently higher at a certain phase, or (2) $\phi_{LF}(t)$ is not uniformly distributed in time. The phase is uniformly distributed in the case of tACS and acoustic stimulation. However, during endogenous sleep, phase is extracted from the slow oscillations, which is not sinusoidal resulting in non-uniform phase distributions. Consequently, we applied a histogram equalization to the phase distribution, ensuring that non-zero modulation index is only a result of modulated amplitude coupling, which we confirmed by testing for significance using surrogate data with constant HF amplitude.

**Randomized surrogate data to estimate statistical significance of PAC**. Significance was determined by randomizing the phase of each LF cycle and thus creating the distribution of the modulation index $r$, under the null hypothesis of no PAC. Phase randomization makes no assumptions on the distribution of the data, except that the phase is uniform, which has been addressed in the paragraph above. For each slow oscillation cycle the low-frequency phase, $\phi_{LF}(t)$, was incremented by a random value uniformly distributed between 0° and 180°. The phase was shifted by the same random phase for all electrodes but independently for different cycles.

The randomizing procedure was repeated 10,000 times and the *p*-value was measured as the rate of a random phase having a modulation index higher than the original data. The minimum numerical *p*-value possible, given the number of randomizations, was $10^{-4}$ (1/number of shuffles). All the *p*-values were corrected for multiple comparisons across electrodes using FDR correction[76] with $q < 0.05$. No correction was performed across segments (as we use segments to determine how reliable a potential entrainment is over longer periods of time) or frequency bands (as these were planned comparison). Uncorrected *p*-values for electrodes that had $p < 0.05$ in at least one stimulation block are shown in Supplementary Figs. 7–9 for tACS, endogenous slow-wave (no stimulation), and acoustic stimulation, respectively.

Note that we examine the acute effects of modulation within a single cycle, thus permitting many single "trials" ($N = 300$ cycles for 1 Hz and $N = 225$ for 0.75 Hz within each 5-min block, repeated 2–6 times for each stimulation intensity; see number of blocks tested for each subject in Supplementary Table 1). This large number of trials permits, in principle, detection of small changes in power within a cycle, on the order of a few percent (e.g., assuming independent noise: sqrt(1/300) = 5.6%). See Supplementary Note 3 for an extended discussion on statistical power.

To aid comparison with endogenous sleep and tACS stimulation condition, we analyzed entrainment for different sleep duration segments. The longer the recordings (approximately 20 min), the more accurate the mean vector strength technique is at measuring entrainment (approximately 50% of electrodes entrained). For shorter durations of endogenous sleep (5 min of iEEG) comparable to the duration of tACS blocks, there is still modulation of spindle activity but in only 10% of the electrodes.

**Spindle event detection**. Spindle detection follows existing methods[50]. Briefly, the signal is bandpass filtered in the spindle band as above, and instantaneous power is obtained as the square amplitude of the filtered signal. For each channel a detection threshold is defined as six times the median of the instantaneous power. For segments that cross this threshold a lower threshold (1 standard deviation of the power) is applied to detect onset and offset of the spindle event. Only spindles whose duration was between 0.2 and 2 s were considered for further analysis[50, 77]. To avoid false positive detections due to patient movements or high-frequency artifacts spindle events that coincide with increases in broadband power were discarded. Events exhibiting broadband power increases ($p > 0.10$, comparing the maximum broadband power in that event vs. the distribution of broadband power during the whole recording) were excluded.

**Slow-wave detection during auditory stimulation**. Additional analyses were performed on the acoustic stimulation data to assess the physiological origin of the PAC entrainment. First, a slow-wave detection algorithm was applied during stimulation segments to assess whether underlying slow-wave activity was altered during stimulation. In contrast to tACS, this is possible for auditory stimulation as there are no electric stimulation artefacts in the iEEG signal. Following previous literature[61, 78], slow-wave detection consisted of bandpass filtering the waveform ($fc = 0.75$ Hz, bandwidth = 0.75 Hz) and using a zero-crossing algorithm to identify events where two subsequent negative zero-crossing (e.g., from positive to negative) were within the range of 0.5–2 s (2–0.5 Hz, bandwidth = 1.5 Hz). Subsequently, once a slow-wave event was identified, the down-state was identified as the minimum voltage within this event. Second, time-frequency response (TFR) functions were calculated to assess whether stimulation trials consisted of physiological sleep spindles. TFRs were computed using a 6-cycle Morlet wavelets between 5 and 25 Hz and averaging the resulting spectrograms (locked to stimulation onset) across all noise bursts (Fig. 6). Significance between conditions (sleep vs. baseline) was assessed via paired sample *t*-test (one-tailed, $p < 0.05$) compared across all stimulation trials ($n = 300$).

**Analysis of changes in power before and after stimulation**. Qualitative comparison between the NREM during the two nights without stimulation and the stimulation period show no evident changes in the power spectrum (Supplementary Fig. 6). To determine if there were significant changes in power at the different frequency bands before vs. after stimulation we used the Chronux toolbox[79] (http://chronux.org/; version 2.12). Briefly, for each electrode, 30 s immediately after each 5-min stimulation block were compared against 30 s preceding the first block. Differences in power in the slow oscillation (0.5–1 Hz) and spindle bands (10 and 14 Hz) are shown in Supplementary Fig. 12. Statistical significance of power changes is computed with the Chronux toolbox for the fast spindle band and slow oscillation band and FDR corrected ($q < 0.05$).

**Entrainment with electric stimulation assessed after stimulation block**. In subject S14 we collected enough stimulation blocks to evaluate slow-wave entrainment in the intervals immediately following tACS. For this, slow-wave entrainment was tested following previous reports[37]. Briefly, the 10-s stimulation-free interval immediately after stimulation was fit with a sinusoid (0.5–1.5 Hz) and phase coherence was calculated across all trials. Statistical significance was calculated using a circular test statistic (Rayleigh test for non-uniformity). In addition, to test for after-effects, we utilized the same trapezoidal waveform, frequency, and intensity in five separate 5-min blocks during NREM sleep (5 min ON/5 min OFF).

We compared these sessions against two control periods of endogenous NREM sleep for spindle (10, 14 Hz) entrainment. The power in the stimulation-free intervals was calculated for slow oscillations (0.5–1 Hz), slow spindle activity (8–12 Hz), and fast spindle activity (12–15 Hz). This quantity was then compared to the power during a different night, when the subject was in an equal sleep state. Slow oscillations and slow spindle activity were averaged across locations close to Fz. And fast spindle activity was determined across parietal locations.

**Code availability**. The code used to generate the main findings of the current study are available from the corresponding author upon reasonable request.

**Data availability**. The data sets generated during and/or analyzed during the current study are available from the corresponding author upon reasonable request.

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

## Acknowledgements

This work was supported by R01 MH107396-01, R44 NS092144, R01 MH-092926, NS095123, R41 NS076123, NYU Program Project Grant Development Initiative (PPG-DI), NYU Clinical and Translational Science Institute (CTSI), NYU Finding a Cure for Epilepsy (FACES), and the Zimin Foundation. The authors would also like to thank Xiuyuan (Hugh) Wang for his work on the electrode reconstruction, as well as Preet Minhas for her early work in data collection and patient coordination.

## Author contributions

Conception and design: G.B., O.D., A.A.L., D.F., L.C.P.; institutional review board approval: A.A.L., T.T., L.M.; acquisition of data: A.A.L., B.L., W.D., D.F., S.H.; analysis and interpretation of data: B.L., Y.H., L.C.P., A.A.L., D.F., G.B., O.D., S.H.; figure preparation: B.L., Y.H., L.C.P., S.H.; drafting the manuscript: A.A.L., L.P., G.B., O.D.; critical revisions: all authors.

## Additional information

**Competing interests:** L.P. has shares in Soterix Medical Devices. The remaining authors declare no competing financial interests.

