## [Peer Review File · Nature Communications]

REVIEWERS' COMMENTS:

Reviewer #1 (Remarks to the Author):

Review of Lafon et al revised ms

I reviewed in detail the revised manuscript by Lafon and colleagues. The authors thoroughly addressed the concerns raised by my original review, as well as the concerns raised by the other referees. The revised manuscript has been substantially improved in the process.

Specifically, the authors satisfactorily address my concerns regarding the statistical power of null effects (5-minute interval issue), multiple stimulation parameters (frequency, intensity, montage), analysis of artifacts, more definitive proof that TACS stimulation was effective, description of the FEM computational model, replicating more closely the original protocol that was reported to be effective, sleep scoring, and analyzing slow spindle activity

The authors rightly state that this additional work strengthens their conclusions, thus increasing the impact and generalizability to the field of transcranial electrical stimulation.

I have one minor comment (below) and I recommend publication at Nature Communications once this issue is addressed.

Minor comment:

- In the additional patient S014 (bilateral electrodes on F3/F4 stimulating during NREM sleep) "The trapezoidal waveform does not permit analysis during stimulation due to the broad band artifacts they generate. Thus, we analyzed whether the slow-wave activity immediately after stimulation remained time--aligned across trials following previous methods"; in my view whether stimulation has sustained effects immediately after stimulation has ended is a separate question. Why not analyze spindle power as was done for other patients? Did the trapezoidal waveform preclude such analysis?

Reviewer #2 (Remarks to the Author):

The authors' response represents a tour de force addressing most of the issues raised by the reviewers in a detailed way.

My fundamental concern remains the paper's message that tACS does not induce biological

effects, when this was tested at a limited number of parameters and locations. In this way the results are over interpreted.

Other than that fundamental point, the study is important, methodologically sound and impactful for the brain stimulation community

Reviewer #3 (Remarks to the Author):

The authors have satisfactorily dealt with all of my points, (except for one minor point, see below). In my view the ms is ready to be published.

To address my previous point 3.1 the authors should explicitly state >> in the ms<< the suggested possibility that the effects reported in Marshall et al. 2006 may result from the DC offset (as in Marshall et al., 2004). This would be a true transcranial brain stimulation effect, just not based on the mechanism of entrainment.

Response to Reviewer Comments NCOMMS-17-13154-T

The authors wish to thank the reviewers for their favorable assessment and remaining comments which are addressed below.

REVIEWER 1

I reviewed in detail the revised manuscript by Lafon and colleagues. The authors thoroughly addressed the concerns raised by my original review, as well as the concerns raised by the other referees. The revised manuscript has been substantially improved in the process.

Specifically, the authors satisfactorily address my concerns regarding the statistical power of null effects (5-minute interval issue), multiple stimulation parameters (frequency, intensity, montage), analysis of artifacts, more definitive proof that TACS stimulation was effective, description of the FEM computational model, replicating more closely the original protocol that was reported to be effective, sleep scoring, and analyzing slow spindle activity

The authors rightly state that this additional work strengthens their conclusions, thus increasing the impact and generalizability to the field of transcranial electrical stimulation.

I have one minor comment (below) and I recommend publication at Nature Communications once this issue is addressed.

Minor comment:

• **In the additional patient S014 (bilateral electrodes on F3/F4 stimulating during NREM sleep) “The trapezoidal waveform does not permit analysis during stimulation due to the broad band artifacts they generate. Thus, we analyzed whether the slow-wave activity immediately after stimulation remained time--aligned across trials following previous methods”; in my view whether stimulation has sustained effects immediately after stimulation has ended is a separate question. Why not analyze spindle power as was done for other patients? Did the trapezoidal waveform preclude such analysis?**

Yes, the trapezoidal waveform creates a broadband artifact which precludes analysis of spindle activity or slow-waves during stimulation. The choice of sinusoidal stimulation for the vast majority of the experiments was motivated precisely by this problem, as the sinusoidal artifact and its harmonics can be cleanly subtracted from the EEG signal, allowing for concurrent spindle analysis. Thus, we were only able to analyze whether slow-wave activity was entrained immediately after each trapezoidal stimulation block, comparable to previously published analyses. We have added the following text to the main body of the manuscript on page 7, paragraph 2:

“The trapezoidal waveform does not permit analysis during stimulation due to the broad band artifacts they generate (for this reason we used sinusoidal stimulation for the majority of the

experiments).”

REVIEWER 2 (Remarks to the Author):

The authors' response represents a tour de force addressing most of the issues raised by the reviewers in a detailed way.

My fundamental concern remains the paper's message that tACS does not induce biological effects, when this was tested at a limited number of parameters and locations. In this way the results are over interpreted.

Other than that fundamental point, the study is important, methodologically sound and impactful for the brain stimulation community.

The authors have been careful to state the limitations of the generalizability of their findings, in several statements made in the discussion:

“Our results therefore do not rule out more sensitive resonant effects of tACS for more rhythmic narrow-band activity such as alpha and spindle oscillations.” (page 12, third paragraph)

“Third, we only considered low-frequency stimulation. Given the dependence of oscillatory stimulation of the specific of network rhythms⁶⁴, these findings may not generalize to other tACS frequencies. In total, as with all null results, one cannot rule out effects outside the parameters tested here.” (page 13, second paragraph)

“We emphasize that our null results for tACS do not contradict the reported behavioral effects. While positive behavioral results have been found in rodents²⁴, a meta-analysis on memory effects reports mixed results in humans³⁹. There may be other means by which non-invasive stimulation affects brain activity in multiple indirect ways, including activation of afferent nerves^{70, 73, 74}, retina and the vestibular apparatus^{75, 76, 77}, astrocytes and perivascular elements^{78, 79}, glial activation⁷⁹, synaptic plasticity, as well as through placebo effects^{80, 81}, which merit further exploration.” (page 14, second paragraph)

REVIEWER 3 (Remarks to the Author):

The authors have satisfactorily dealt with all of my points, (except for one minor point, see below). In my view the ms is ready to be published.

To address my previous point 3.1 the authors should explicitly state >> in the ms<< the suggested possibility that the effects reported in Marshall et al. 2006 may result from the DC offset (as in Marshall et al., 2004). This would be a true transcranial brain stimulation effect, just not based on the mechanism of entrainment.

We agree, the net DC current may be the actual driver of effects in earlier studies and would not rely on entrainment effects. We have added the following text: “In total, as with all null results, one cannot rule out effects outside the parameters tested here. In particular, it is possible that there were lasting effects on spindle power, which we did not resolve here because of their natural fluctuations during sleep on the time scales of minutes. Such changes were reported in past studies^{16,21} by averaging over many subjects and may have resulted from the net-DC currents used there, which we did not test here.”